# Comparison of the Treatment Efficacy of Rosuvastatin versus Atorvastatin Loading Prior to Percutaneous Coronary Intervention in ST-Segment Elevation Myocardial Infarction

**DOI:** 10.3390/jcm11175142

**Published:** 2022-08-31

**Authors:** Esraa M. Adel, Ahmed A. Elberry, Ahmed Abdel Aziz, Ibrahim A. Naguib, Badrah S. Alghamdi, Raghda R. S. Hussein

**Affiliations:** 1National Heart Institute, Cairo 12461, Egypt; 2Department of Pharmacy Practice, Pharmacy Program, Batterjee Medical College, Jeddah 21442, Saudi Arabia; 3Department of Clinical Pharmacology, Faculty of Medicine, Beni-Suef University, Beni-Suef 62513, Egypt; 4Department of Pharmaceutical Chemistry, College of Pharmacy, Taif University, Taif 21944, Saudi Arabia; 5Neuroscience Unit, Department of Physiology, Faculty of Medicine, King Abdulaziz University, Jeddah 22252, Saudi Arabia; 6Pre-Clinical Research Unit, King Fahd Medical Research Center, King Abdulaziz University, Jeddah 21589, Saudi Arabia; 7Department of Clinical Pharmacy, Faculty of Pharmacy, Beni-Suef University, Beni-Suef 62514, Egypt; 8Department of Clinical Pharmacy, Faculty of Pharmacy, October 6 University, 6 October City 12573, Egypt

**Keywords:** hydrophilic, lipophilic, atorvastatin, rosuvastatin, statin, STEMI, PCI

## Abstract

The aim of this study was to compare the effect of a single high-dose rosuvastatin versus atorvastatin preloading in ST-elevation myocardial infarction (STEMI) patients receiving primary percutaneous coronary intervention (PCI.) Methods: A total of 99 patients presented with STEMI and were randomly divided into three groups—a control group (*n* = 33) with no statin treatment, an atorvastatin group (*n* = 33) with a single 80 mg atorvastatin dose and the rosuvastatin group (*n* = 33) with a single 40 mg rosuvastatin dose in the emergency room (ER) prior to PCI. Post-interventional thrombolysis in myocardial infarction (TIMI) flow grade and corrected TIMI frame count (CTFC) were recorded, and ST-segment resolution was measured. Results: CTFC was significantly lower for the atorvastatin group (*p*-value < 0.01) than in the control group. A final TIMI flow grade 3 was achieved in 32 (97.0%) patients in the rosuvastatin group and 28 (84.8%) patients in the atorvastatin group compared with only 25 (75.8%) patients in the control group (*p* = 0.014). Peak CK-MB in the rosuvastatin group (263.2 [207.2–315.6]) and the atorvastatin group (208 [151.0–314.1]) was lower compared to that in the control group (398.4 [303.9–459.3]); *p* < 0.001. Conclusions: A single extensive dose of lipophilic atorvastatin prior to primary PCI in STEMI patients showed better improvement in microvascular myocardial perfusion compared to hydrophilic rosuvastatin.

## 1. Introduction

In many STEMI cases, after primary PCI, myocardial tissue cannot be perfused despite restoration of blood flow within infarct-related arteries—this is called the no-reflow phenomenon [1,2]. Although the pathophysiology of no-reflow is poorly understood, it is supposed to occur through several factors involving microdistal embolization and ischemia–reperfusion injury [3,4,5], which is a predictor of future myocardium remodeling and diminished cardiac function [5,6]. Many studies performed on STEMI patients ending with previous use of statins may improve coronary blood flow after PCI [5,7,8]. Statins have beneficial effects on the vascular system through non-lipid mechanisms such as providing positive actions on platelet adherence, thrombosis, endothelial function, stability of plaque, and inflammation, which are called pleiotropic effects [9,10,11]. Little data exist to compare single high-dose preloading of rosuvastatin (hydrophilic statin) versus atorvastatin (lipophilic statin) for improving coronary blood flow in the event of ST-elevation myocardial infarction (STEMI) early after the primary PCI. For this reason, the present study is designed to conduct those comparative results.

## 2. Patients and Methods

Study population: The current study was a prospective, randomized controlled trial (RCT) with a closed envelope method on 99 Egyptian patients with STEMI conducted at the national heart institute. The current study was registered prospectively at Clinical-Trials.gov (ID: NCT04974814), and the date of registration was 22 July 2021.

Inclusion criteria included the occurrence of symptoms within 12 h, ST-segment elevation ≥ 0.1 mV in at least two contiguous leads of electrocardiogram (ECG) [12], and age of patients ranged from 18 to 80 years. Exclusion criteria were pre-existing (within 3 months) or current statin treatment; allergy to any medication used in hospital; pregnancy; and cardiogenic shock. The primary selection of patients included 155 patients. Fifty six patients were excluded from this study because of referral to fibrinolytic therapy rather than primary PCI; hence, 99 eligible patients were randomly divided into 3 groups—a control group (*n* = 33) with no statin treatment, an atorvastatin group (*n* = 33) with a single 80 mg atorvastatin dose [13] and a rosuvastatin group (*n* = 33) with a single 40 mg rosuvastatin dose [14] in ER prior to PCI. All enrolled patients received guidelines—recommended therapy before and after PCI independent from previous randomized assignment including the usual maintenance statin dose [12,15]. The physicians conducting the intervention and follow-up evaluations were blind to the randomization assignment.

Angiographic, electrocardiographic and enzymatic analysis: Angiograms were reviewed before and after PCI.

The thrombolysis in myocardial infarction (TIMI) flow grade [16] pre- and post-PCI was determined. TIMI flow grade was assessed based on a scale of 0 to 3; TIMI 0 (no perfusion): total occlusion, TIMI 1 (penetration without perfusion): contrast penetration of obstruction without distal perfusion, TIMI 2 (partial perfusion): perfusion through the whole artery but with slow flow, and TIMI 3 (complete perfusion): total perfusion with normal flow.

Corrected TIMI frame count (CTFC) after PCI [3] was examined, defined as: counting the number of frames required for the dye to reach the standardized distal coronary landmark in an infarct-related artery. A correction factor was required for the compensation of the longer length of the left anterior descending artery (LAD) compared with the left circumflex (LCX) and the right coronary artery (RCA), hence the number of frames required for the dye to penetrate the LAD was divided by 1.7.

The following distal landmark branches were used for analysis: the distal bifurcation of the LAD; in the circumflex, the latest branch off most distal OM that included the culprit lesion with the longest total distance and in RCA; the first branch of the posterolateral artery. Frame rate must be adjusted to 30 frames/s (i.e., if the rate was 15 frames/s, the frame count was multiplied by a factor of 2). CTFC and TIMI flow grade were detected by experienced observers who were blinded to group randomization as described previously. CTFC was evaluated by the same experienced observer for all individuals and, based on previous studies, CTFC is non-significantly different among observers or even within the same observer [17,18,19,20,21]. 

The ECGs before and 90 min after primary PCI were collected for analysis by skilled physicians who were blinded to group randomization using a Mindray electrocardiograph (model BeneHeart R3). ST-segment resolution (STR) was measured [22] as the percentage of the summed ST-segment elevations on initial ECG minus the summed ST-segment elevations on the ECG at 90 min after PCI, divided by the summed ST-segment elevations on initial ECG. The complete STR was defined as 70% STR. Blood samples for laboratory cardiac enzymes analysis were collected as much as possible before and after PCI.

Statistical Analysis:

Sample Size Calculation

The necessary sample size has been calculated using the G*Power software [Computer software], version 3.1.7 [23,24]. The main outcome parameter is the difference among the three groups in terms of the CTFC. No previous study has compared the effect of preloading with rosuvastatin or atorvastatin on CTFC in patients with ST-segment elevation myocardial infarction (STEMI) undergoing PCI. So, we targeted an effect size that would be clinically relevant. We calculated that a sample size of 33 patients for each group achieves 80% power to identify a statistically significant difference among the 3 groups in terms of the CTFC when the difference is medium sized, corresponding to a *Cohen’s f* of 0.3. We used a two-sized *F-test* with numerator and denominator degrees of freedom of 2 and 96, respectively, and targeted the test significance at the *p* < 0.05 level (Critical F = 3.091). We chose a medium effect size (*Cohen’s f*) of 0.3 as we considered it to be clinically important. *Cohen’s f* is calculated as follows:$$Cohen^{\prime }s\;f=\frac{Between\;group\;SD}{Within\;group\;SD}$$
where *SD* is the standard deviation.

Data were determined using IBM© SPSS© Statistics version 26 (IBM© Corp., Armonk, NY, USA). Normally distributed numerical variables are presented as the mean and SD and inter-group differences were compared using one-way analysis of variance (ANOVA). Skewed numerical data are presented as the median and interquartile range and differences were compared using the Kruskal–Wallis test with application of the Conover post hoc test if needed. Categorical variables are presented as ratios or numbers and percentages and differences were compared using the Pearson chi-square test or Fisher’s exact test as appropriate. Ordinal data were compared using the chi-square test for trend Multivariable linear (or binary logistic) regression analysis is used to determine the effect of the interventions on the main outcome measures as adjusted for the effect of other confounding factors. Skewed numerical variables were subjected to logarithmic transformation prior to entry in linear regression. For one-way comparisons, *p*-values < 0.05 are considered statistically significant. For post hoc pairwise comparison with the Conover test, we applied the Bonferroni method to adjust the critical *p*-value for the number of comparisons, which indicated that the *p*-value should be <0.0167 to be considered statistically significant.

## 3. Results

### 3.1. Baseline Characteristics

The basic clinical features of the three groups examined are displayed in Table 1. Age, sex and onset to presentation time insignificantly differed among the three groups; also the difference was not significant regarding risk factors for CAD except for smoking in the rosuvastatin group 27 (81.8%) versus only 17 (51.5%) in the atorvastatin group and 18 (54.5%) in the control group (*p* = 0.02). Coronary angiographic findings are displayed in Table 2. The culprit lesion in the three study groups insignificantly differed from each other (*p* = 0.66), and TIMI flow grade before performing PCI also insignificantly differed between groups (*p* = 0.90).

### 3.2. Angiographic, Electrocardiographic, Echocardiographic and Enzymatic Findings

#### 3.2.1. TIMI Flow Grade 

After PCI, as shown in Figure 1, 97% of patients in the rosuvastatin group and 84.8% of patients in the atorvastatin group had TIMI flow grade III compared with only 75.8% of patients in the control group. Differences among the three groups are statistically significant (*p*-value = 0.014).

#### 3.2.2. CTFC 

After PCI, as illustrated in Figure 2, the difference was statistically significant between the atorvastatin and control groups (*p*-value < 0.01). Differences between the rosuvastatin and atorvastatin groups and between the rosuvastatin and control groups were not statistically significant.

#### 3.2.3. STR 

After PCI, as shown in Figure 3, 39.4% of patients in both the rosuvastatin and atorvastatin groups had complete STR compared with only 15.2% of patients in the control group. Differences among the three groups are only marginally significant statistically (*p*-value = 0.0495).

#### 3.2.4. Peak CK-MB 

After PCI, as shown in Figure 4, differences between the rosuvastatin and control groups and between the atorvastatin and control groups were statistically significant (*p*-values < 0.01). The difference between the rosuvastatin and atorvastatin groups was not statistically significant.

#### 3.2.5. LVEF 

At hospital discharge, as illustrated in Table 3, after adjustment for the effect of onset to presentation time, both the rosuvastatin (regression coefficient = 0.054, SE = 0.019, *p*-value = 0.005) and atorvastatin (regression coefficient = 0.052, SE = 0.019, *p*-value = 0.007) groups were significantly associated with higher LVEF.

## 4. Discussion

This comparative study showed that a single high-dose rosuvastatin and atorvastatin loading before primary PCI was associated with significantly higher TIMI flow grade and significantly lower peak CK-MB. Both the rosuvastatin and atorvastatin groups were predictors of complete STR. CTFC was statistically significantly lower in the atorvastatin group than in the control group, but the rosuvastatin group failed to achieve that significance. 

Although TIMI flow grade, STR, and CTFC are usually used in the assessment of microvascular coronary perfusion, half of the patients with post-procedural normal TIMI 3 flow achievement visually, distal microvascular coronary perfusion is not fully restored leading to poor clinically outcomes. Accordingly, instead of limitations of TIMI flow grading including interobserver variability and non-quantitative nature, CTFC introduced a sensitive, reproducible, simple, and quantitative tool to accurately assess coronary microcirculation blood flow which represents a huge portion of myocardium blood supply [17,25]. Additionally, STR is demonstrated to represent an inaccurate method to determine microvascular coronary perfusion because of less sensitivity in non-LAD than LAD infarction and its influence on collateral circulation [26,27,28,29]. Finally, although peak CK-MB is considered the most powerful enzymatic predictor of infarct size [30,31], the infarct size is not affected by the no-reflow incidence alone. In addition to the no-reflow impact on the infarct size, there are also several factors that could affect it such as ischemic duration and collateral circulation [32].

Many studies demonstrated that high-dose atorvastatin pretreatment reduces periprocedural MI and myocardial injury in both stable angina during elective PCI and ACS patients undergoing an early invasive strategy [33,34,35,36].

The first trial performed to study the effectiveness of a single high-dose atorvastatin loading in STEMI patients immediately before primary PCI was the STATIN-STEMI study in which a comparison was performed between 80 mg atorvastatin and 10 mg atorvastatin loading in ER compared to conventional treatment. CTFC was significantly lower and complete STR was significantly higher in the 80 mg atorvastatin group, but in contrast to the current study, the peak CK-MB did not differ among groups. The STATIN-STEMI ended with a high dose of atorvastatin pretreatment improved microvascular coronary perfusion [37].

After that, Kim et al. [38] assessed the effect of high-dose rosuvastatin (40 mg) loading also in STEMI patients prior to primary PCI on infarct size, which was detected with single-photon emission computed tomography (SPECT). The infarct size was assessed by SPECT and serial cardiac biomarkers measurements—both were significantly lower in the rosuvastatin group than in the control group; further, improved coronary microvascular perfusion, assessed by CTFC, was found in the rosuvastatin group. The multivariate analysis revealed that rosuvastatin loading, pain to balloon time (OR 2.05), anterior myocardial infarction (OR 3.89) and final MBG (OR 2.93) were independent predictors of large infarct size. However, unlike the results of the current study, CTFC was significantly lower in the rosuvastatin group and LVEF did not differ between the two studied groups, but these findings disagreed with the ROSEMARY study [39], in which 185 patients with STEMI undergoing primary PCI were divided to either the high-dose rosuvastatin group (rosuvastatin 40 mg preloading then maintenance for 7 days) or the conventional low-dose rosuvastatin group (placebo preloading then 10 mg maintenance for 7 days). Series cardiac magnetic resonance imaging (CMR) was carried out during acute (3 to 7 days) and chronic (3 months) phases to investigate whether high-dose rosuvastatin pretreatment can reduce infarct size compared with the conventional low dose. The results revealed that the relative infarct volumes in the acute and chronic phases were not different between the groups. Microvascular coronary circulation evaluated by TIMI flow grade, MBG, STR and microvascular obstruction on CMR showed no difference between groups, hence high-dose rosuvastatin pretreatment did not improve periprocedural myocardial perfusion or reduce the volume of infarction measured by CMR. Findings from the ROSEMARY study might be more reliable as CMR is known as the gold-standard diagnostic method for assessment of microvascular coronary perfusion infarct size compared with SPECT, the accuracy of which is influenced by the stunning myocardium [27,40,41,42].

Garcia et al. [43] demonstrated that an extensive dose of atorvastatin administration prior to primary PCI may improve microvascular coronary perfusion assessed by combined no-reflow parameters (ST < 50%, no-reflow angiographically and no-reflow by SPECT). In the study, 103 STEMI patients within 12 h duration of symptoms received either 80 mg atorvastatin additional to standard treatment (AST) before undergoing primary PCI or standard treatment alone (ST). The frequency of no-reflow among groups was 27% vs. 63%, respectively, and the multivariate analysis showed that the treatment assigned (80 mg atorvastatin preloading) was the only independent predictor for the combined no-reflow occurrence.

A meta-analysis performed on fifteen randomized controlled trials revealed that high-dose statin administration before PCI resulted in a significant improvement in coronary blood flow assessed by post-procedure TIMI flow grade in comparison to the control group [44]. In this study, analysis of atorvastatin subgroup demonstrated that high-dose atorvastatin prior to PCI significantly affects TIMI flow grade after PCI; however, no significant effect was observed in the rosuvastatin subgroup analysis.

All these positive effects on coronary blood flow and myocardial perfusion were achieved because statins have non-lipid-lowering mechanisms called pleiotropic effects, which include plaque stability, inhibition of platelet aggregation, anti-thrombotic and anti-inflammatory actions and enhancement of endothelial function [45,46,47,48]. The difference in the pleiotropic effects between lipophilic (atorvastatin) and hydrophilic statins is thought to be because lipophilic statins are widely distributed within extra-hepatic tissues, while hydrophilic statins are hepatoselective and are thus suggested to have less pleiotropic efficacy [49]. Furthermore, hydrophilic statins were found to have very poor myocardium uptake in contrast to lipophilic statins, which are highly taken up by cardiac muscles [50]. Finally, concern about allowing enough time for statins to exert their pleiotropic efficacy might be answered by previous studies. Bell et al. [51] demonstrated that only ten minutes of preloading with atorvastatin could inhibit myocardial injury resulting from procedural reperfusion.

Study limitations: First, the sample size in this study is relatively small, so further studies with a larger population are recommended to confirm comparison results between the two types of statins studied. Second, follow up was only performed for the duration of hospital stay of patients, so long-duration follow up to identify the effect of statins on myocardium function improvement after acute phase or at 1 year follow up for testing their effect on reducing MACEs (major adverse cardiac events). In addition, the external validity of the current study may be limited as the present study is only applicable to Egyptian patients. Finally, the exact mechanisms of the pleiotropic effects of statins need further investigation as they are not fully explained. 

## 5. Conclusions

Single high-dose atorvastatin pretreatment before primary PCI in statin naïve STEMI patients might be superior to high-dose rosuvastatin preloading in improving microvascular coronary perfusion right after PCI.

## Figures and Tables

**Figure 1 jcm-11-05142-f001:**
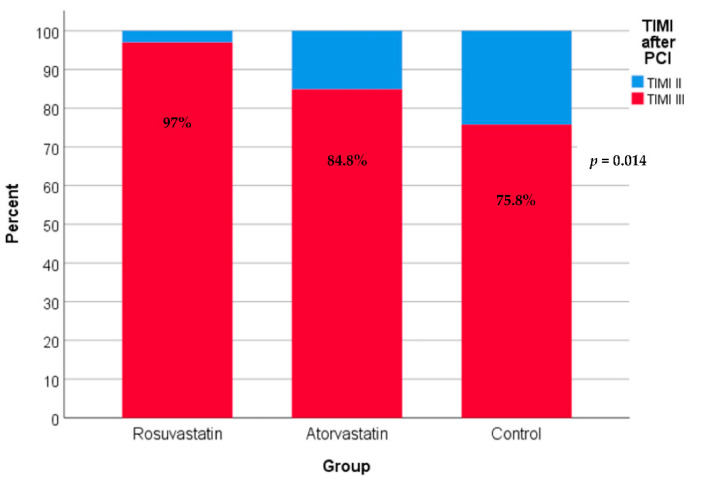
TIMI (thrombolysis in myocardial infarction) flow grade after PCI in the three study groups. Using the chi-square test for trend.

**Figure 2 jcm-11-05142-f002:**
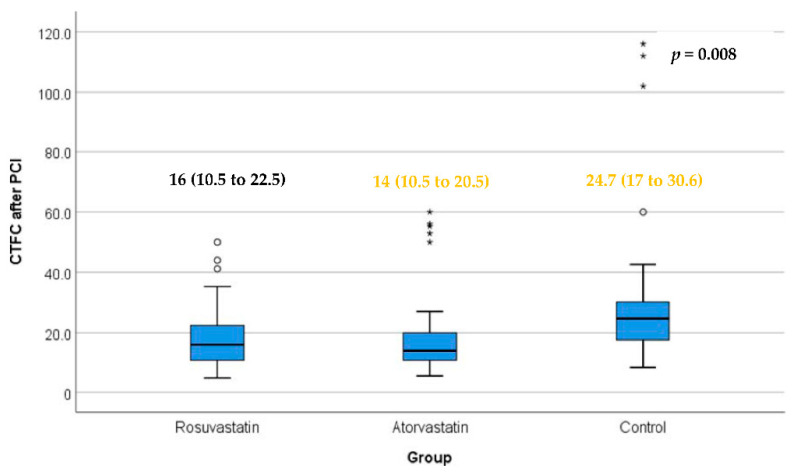
CTFC (corrected TIMI frame count) after PCI in the three study groups. Box represents the interquartile range. Line inside the box represents the median. Error bars represent minimum and maximum values excluding outliers (rounded markers) and extreme values (asterisks). There was a statistically significant difference between the atorvastatin and control groups. Differences between the rosuvastatin and atorvastatin groups and between the rosuvastatin and control groups were not statistically significant. *p* = 0.003 for atorvastatin versus control (Conover test); *p* = 0.020 for rosuvastatin versus control; *p* = 0.788 rosuvastatin versus atorvastatin (Conover test).

**Figure 3 jcm-11-05142-f003:**
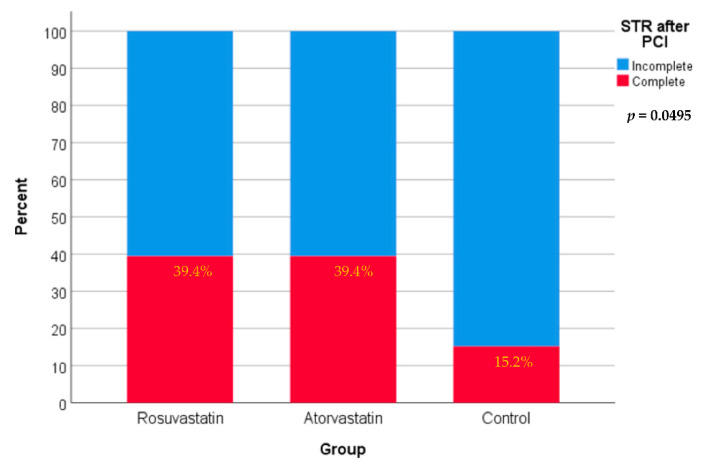
STR (ST-segment resolution) after PCI in the three study groups using the Pearson chi-square test.

**Figure 4 jcm-11-05142-f004:**
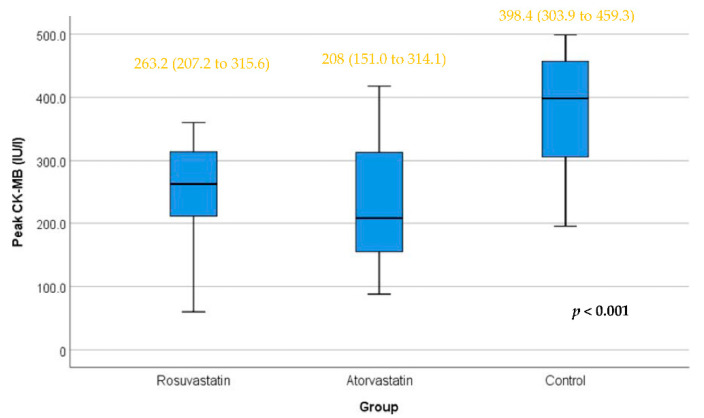
Peak CK-MB level in the three study groups. Box represents the interquartile range. Line inside the box represents the median. Error bars represent minimum and maximum values. Differences between the rosuvastatin and control groups and between the atorvastatin and control groups were statistically significant (*p*-values < 0.01). The difference between the rosuvastatin and atorvastatin groups was not statistically significant. The test applied was the Kruskal–Wallis test.

**Table 1 jcm-11-05142-t001:** Baseline characteristics of the three study groups.

Variable	Rosuvastatin (*n* = 33)	Atorvastatin (*n* = 33)	Control (*n* = 33)	*p*-Value
Age (years), mean ± SD	55.4 ± 8.7	53.2 ± 9.9	57.7 ± 7.6	0.122 †
Sex, F/M	5/28	7/26	8/25	0.645 ‡
BMI (kg/m^2^), mean ± SD	29.6 ± 5.3	27.9 ± 4.6	29.6 ± 6.5	0.328 †
Family history of CAD, *n* (%)	7 (21.2%)	6 (18.2%)	9 (27.3%)	0.664 ‡
Hypertension, *n* (%)	10 (30.3%)	14 (42.4%)	14 (42.4%)	0.505 ‡
DM, *n* (%)	13 (39.4%)	15 (45.5%)	16 (48.5%)	0.751 ‡
Smoking, *n* (%)	27 (81.8%)	17 (51.5%)	18 (54.5%)	0.020 ‡
Onset to presentation time (h), median (IQR)	6.0 (4.0 to 8.0)	7.0 (5.0 to 9.25)	7.0 (4.0 to 8.25)	0.279 §

SD = standard deviation; *n* = number. F = female; M = male; BMI = body mass index. CAD = coronary artery disease; DM = diabetes mellitus. h = hour; IQR = interquartile range. † One-way analysis of variance. ‡ Chi-square test for trend. § Kruskal–Wallis test.

**Table 2 jcm-11-05142-t002:** Coronary angiographic findings in the three study groups.

Variable	Rosuvastatin (*n* = 33)	Atorvastatin (*n* = 33)	Control (*n* = 33)	*p*-Value
Culprit lesion				0.660 †
*RCA, n (%)*	8 (24.2%)	10 (30.3%)	5 (15.2%)	
*LAD, n (%)*	23 (69.7%)	22 (66.7%)	26 (78.8%)	
*LCX, n (%)*	2 (6.1%)	1 (3.0%)	2 (6.1%)	
TIMI before PCI				0.906 ‡
*TIMI 0, n (%)*	27 (81.8%)	22 (66.7%)	28 (84.8%)	
*TIMI I, n (%)*	2 (6.1%)	1 (3.0%)	0 (0.0%)	
*TIMI II, n (%)*	2 (6.1%)	2 (6.1%)	4 (12.1%)	
*TIMI III, n (%)*	2 (6.1%)	8 (24.2%)	1 (3.0%)	

*n* = number. RCA = right coronary artery. LAD = left anterior descending. LCX = left circumflex. TIMI = thrombolysis in myocardial infarction. † Fisher’s exact test. ‡ Chi-square test for trend.

**Table 3 jcm-11-05142-t003:** Multivariable linear regression analysis for effect of rosuvastatin or atorvastatin on the LVEF at hospital discharge as adjusted for onset to presentation time.

Independent Variables	Coefficient	Std. Error	t	*p*-Value	r (Partial)	r (Semipartial)
(Constant)	1.666					
Rosuvastatin (=1)	0.054	0.019	2.857	0.005	0.281	0.278
Atorvastatin (=1)	0.052	0.019	2.739	0.007	0.271	0.267
Log onset to presentation time (hours)	0.003	0.035	0.074	0.941	0.008	0.007

Dependent variable: Log LVEF at hospital discharge (%).

## Data Availability

The data will be available from the corresponding author upon request.

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
