# Peer review of "Comparison of the Treatment Efficacy of Rosuvastatin versus Atorvastatin Loading Prior to Percutaneous Coronary Intervention in ST-Segment Elevation Myocardial Infarction"

_jcm, 2022, doi:10.3390/jcm11175142_

Round 1

Reviewer 1 Report

The introduction can be made more succinct to focus on the study question. Please reduce it to a single paragraph that introduces the study question. 

Please report missing values in the dataset. 

Why was multivariable logistic regression used in a RCT?

Logistic regression is described in the study methods, however, i don't see the results from the regression model in the results

How many patients were lost to follow up?

Do we have data on the race/ethnicity of the individuals?

Please provide p values for direct comparison of atorvastatin vs rosuvastatin.

Author Response

Reviewer 1

Comment

Author Response

Text Insertion (if applicable)/ section

The introduction can be made more succinct to focus on the study question. Please reduce it to a single paragraph that introduces the study question.

we briefly conducted the introduction of the manuscript

Introduction

Page 1

Line 36

Please report missing values in the dataset.

We completed the missing parts.

Figure A2

Figure A3

Figure A4

Why was multivariable logistic regression used in a RCT?

Multivariable linear regression analysis was used to determine the effect of the interventions on the main outcome measures as adjusted for the effect of other confounding factors.

Logistic regression is described in the study methods; however, I don’t see the results from the regression model in the results.

Missed logistic regression results have been added as they were lost unintentionally.

Page 4

Line 157

Page 5

Line 175

Page 7

Table 3

How many patients were lost to follow up?

56 patients who were lost to be follow up because of referral to fibrinolytic therapy rather than primary PCI

Page 2

Line 61

Do you have data on the race/ethnicity of the individuals?

All populations included in the current study are Egyptians

Page 2

Line 54

please provide p values for direct comparison of

Atorvastatin vs. rosuvastatin.

done

Figure A2

Reviewer 2 Report

In the present study Dr Adel and coworkers investigated the impact of high statin loading dose in the reperfusion injury after STEMI pPCI. The reperfusion injury was evaluated by TIMI flow, CTFC and ST segment resolution. The authors aimed to compare high atorvastatin and high rosuvastatin loading dose in the ER with controls.

Comments:

1. In the abstract the loading dose of rosuvastatin is not mentioned

2. Some numbers are not clear in the figures

3. In figures 1-3 how was subgroup analysis performed? Which correction was applied? Between group comparisons were performed using ANOVA or Kruskal Wallis?

4. In the statistical analysis section it is mentioned that a mutlivariate linear regression analysis was performed. However, no respective results are mentioned

5. Why was CK-MB and not hs-troponin measured?

6. Did the authors evaluated an interobserver variability for CTFC?

7. How is the conclusion supported by the results of the study? In all three analysis (Figure 1-3) there was no difference between high dose of rosuvastatin or atorvastatin.

Author Response

Reviewer 2

 Comments

Author Response

Text Insertion (if applicable)/ section

1. In the abstract the loading dose of rosuvastatin is not mentioned.

We added the loading dose of rosuvastatin missed in the abstract.

Page 1

Line 24

2. Some numbers are not clear in the figures.

Done

Figure A2

Figure A3

Figure A4

3. In figures 1-3 how was subgroup analysis performed? Which correction was applied? Between groups comparisons were performed using ANOVA or Kruskal Wallis?

done

Page 3

Line 128

Figure A1

Figure A2

Figure A3

Figure A4

4. In the statistical analysis section it is mentioned that a multivariate linear regression analysis was performed. However, no respective results are mentioned

Missed logistic regression results have been added as they were lost unintentionally.

Page 4

Line 157

Page 5

Line 175

Page 7

Table 3

5. Why was CK-MB and not hs-troponin measured?

We used peak CK-MB for enzymatic infarct size evaluation as peak CK-MB is considered the most powerful enzymatic predictor of infarct size [1, 2]. It is not used for early detection and diagnosis of STEMI; furthermore, CK-MB is the best enzyme test used for detection of re-infarction incidence [3-5].

Page 8

Line 225

6. Did the authors evaluate an interobserver variability for CTFC?

No, as it was the same observer who evaluated CTFC for all individuals and he was blinded to the randomization assignment. Besides; the lower incidence of intra-observer and even inter-observer variations and according to previous studies, that variations are non-significantly differ among observers or the same observer [6-9].

Page 2

Line 90

7. How is the conclusion supported by the results of the study? In all three analyses (figure 1-3) there was no difference between high dose of rosuvastatin or atorvastatin.

In figure 2 regarding the CTFC, There is a statistically significant difference between Atorvastatin and Control (p-value =0.003). Differences between Rosuvastatin and Atorvastatin and between Rosuvastatin and Control are not statistically significant. And as a result, that’s may give superiority to atorvastatin over rosuvastatin as atorvastatin could reach significance compared to control, while rosuvastatin failed to achieve that significance improvement in CTFC. Beside; the sensitivity of the CTFC parameter compared to TIMI flow grade and STR.

Figure A2

Round 2

Reviewer 1 Report

The authors have provided responses and improved the manuscript. I have a few follow up questions.

1- In response to my earlier comment to report missing values in the dataset. The authors responded "We completed the missing parts"

This response is not adequate. My questions was about missing value analysis in the datasett?

2-Were there any missing values in any of the variables in the dataset?

3-If there were missing values in any of the variables, was a missing value anlaysis done to see if they are missing at random or not?

4-In the regression analysis, were any variables used that had missing data?

5-The lost to follow up of 55 patient with a study sample size of 99 is a major concern and deserves a specific mention in the limitations section

6-Please mention in limitations that this study is only applicable to Egyption population, and that external validity of the study maybe limited. 

Author Response

Reviewer 1

Comment

Author Response

Text Insertion (if applicable)/ section

1. In response to my earlier comment to report missing values in the dataset. The authors responded “We completed the missing parts”

This response is not adequate. My question was about missing value analysis in the dataset?

We apologize about the previous confused response as there were some invisible numbers on the related figures.

The current study has no missing values for all of the analyzed variables; as we had a complete dataset for all the variables observed related to the entire 99 individuals included in the study.

2. Were there any missing values in any of the variables in the dataset?

3. If there were missing values in any of the variables, was a missing value analysis done to see if they are missing at random or not?

No, there were not any missing values in any of the variables in the dataset.

4. In the regression analysis, were any variables used that had missing data?

No, there were not any variables used in regression analysis that had missing data.

5. The lost to follow up of 55 patients with a study sample size 99 is a major concern and deserve a specific mention in the limitation section

There was a misunderstanding of the previous comment “how many patients were lost to follow up” as the entire 99 eligible individuals of the study totally followed up and had no missing data. The mentioned number of patients (56) was excluded from 155 initially selected patients as they were not eligible for the current study. Briefly, the answer there were no any patients lost to follow up.

Page 2

Line 61

Reviewer 2 Report

The authors have adequately addressed my comments

Author Response

Thanks so much